# Foreign Medical Students in Eastern Europe: Knowledge, Attitudes and Beliefs about Medical Cannabis for Pain Management

**DOI:** 10.3390/ijerph18042137

**Published:** 2021-02-22

**Authors:** Vsevolod Konstantinov, Alexander Reznik, Masood Zangeneh, Valentina Gritsenko, Natallia Khamenka, Vitaly Kalita, Richard Isralowitz

**Affiliations:** 1Department of the General Psychology, Penza State University, 440026 Penza, Russia; 2Regional Alcohol and Drug Abuse Research Center, Ben Gurion University of the Negev, Beer Sheva 8410501, Israel; reznikal@bgu.ac.il (A.R.); richard@bgu.ac.il (R.I.); 3School of Liberal Arts and Sciences, Humber College Institute of Technology and Advanced Learning, Toronto, ON M9W 5L7, Canada; masood.zangeneh@humber.ca; 4Department of Social Psychology, Moscow State University of Psychology and Education, 123290 Moscow, Russia; so@so.mgppu.ru; 5Department of Psychiatry and Medical Psychology, Belarusian State Medical University, 220116 Minsk, Belarus; khamenka3@gmail.com; 6Department of Pedagogy and Psychology of Professional Education, Moscow State University of Technology and Management Named after K.G. Razumovski, 109004 Moscow, Russia; 700200@mail.ru

**Keywords:** medical cannabis, foreign medical students, pain management

## Abstract

Objective: To assess the knowledge, attitudes, and beliefs of foreign students toward the use of medical cannabis (MC) for pain management. Methods: This study uses data collected from 549 foreign students from India (*n* = 289) and Middle Eastern countries mostly from Egypt, Iran, Syria, and Jordan (*n* = 260) studying medicine in Russia and Belarus. Data collected from Russian and Belarusian origin medical students (*n* = 796) were used for comparison purposes. Pearson’s chi-squared and *t*-test were used to analyze the data. Results: Foreign students’ country of origin and gender statuses do not tend to be correlated with medical student responses toward medical cannabis use. Students from Russia and Belarus who identified as secular, compared to those who were religious, reported more positive attitudes toward medical cannabis and policy change. Conclusions: This study is the first to examine the attitudes, knowledge, and beliefs toward medical cannabis among foreign students from India and Middle Eastern countries studying in Russia and Belarus, two countries who oppose its recreational and medicine use. Indian and Middle Eastern students, as a group, tend to be more supportive of MC than their Russian and Belarusian counterparts. These results may be linked to cultural and historical reasons. This study provides useful information for possible medical and allied health curriculum and education purposes.

## 1. Introduction

Cannabis is the most commonly used psychoactive drug worldwide. However, numerous countries maintain a fundamental position against its use grounded in international obligation under the Single Convention on Narcotic Drugs of 1962, the Convention on Psychotropic Substances of 1971, and the United Nations Convention against Illicit Traffic in Narcotic Drugs and Psychotropic Substances of 1988. In countries such as India and Middle Eastern countries, as well as Russia and Belarus, cannabis is subject to restriction and penalty for medical and recreational use. 

Much interest and advocacy is being generated about cannabis with numerous countries preparing to or having already legalized medical cannabis (MC) use. This is, in part, a result of epidemiological studies and randomized controlled trials that have explored cannabis effectiveness for medical conditions such as chronic pain [1]. Pain, especially chronic, is globally recognized as a major cause of disability. Worldwide, 1 in 5 adults is affected by this condition with 1 in 10 adults becoming newly diagnosed each year. The use of medical cannabis for pain management is a promising treatment option [2]. 

In 2019, over 280,000 foreign students from over 100 countries attended universities in Russia and Belarus [3,4]. For the most part, these international students originated from China, India, former USSR countries such as Kazakhstan, Turkmenistan, and Uzbekistan, and countries within Africa and the Middle East. Among these students, about one-fifth (21%) study medicine [4,5]. For foreign students who do not speak Russian, instruction is in English with parallel study in Russian.

Medical education in both Russia and Belarus is considered high quality; taught by experienced academic personnel, studying a curriculum that includes both English and French learning materials, reasonable tuition cost, and the availability of scholarships [6,7]. Many countries have bi-lateral agreements with Russia and Belarus recognizing international student medical education. However, such recognition may include a qualifying exam in the country of origin for the student to practice medicine in their home country [7].

### 1.1. India

Cannabis in India has been used since as early as 2000 BCE. From that time on, extensive knowledge has accumulated about cannabis in the treatment of various diseases [8]. Ayurveda, the traditional system of Indian folk medicine, lists about 30 different diseases for which cannabis and cannabis-based products have been used to treat pain and other medical conditions [9]. Current medical use of cannabis in India is limited by its illegal status [10]. However, in 2020, India’s first cannabis clinic was opened using cannabis-based medications [11].

In addition to its use for medical purposes, bhang (i.e., a popular cannabis edible mixture made from the buds, leaves, and flowers of the female plant) is used for Hindu religious ceremonies and rituals. Despite its illegal status, exception has been made for production and use-related purposes [12]. In 2019, 2.8% of India’s population aged 10 to 75 reported cannabis use [13]. 

### 1.2. Middle Eastern Countries

The Islamic world became acquainted with cannabis 200 years after the death of the Prophet Muhammad (A.D. 570–632) [14]. Unlike alcohol, there is no direct prohibition of cannabis use in the Quran. Although the status of cannabis in Islam remains uncertain, negative attitudes prevail even though it is allowed for pain and other medical conditions [15]. Despite legal bans and religious restrictions, cannabis use remains present in many Arab countries, with some like Lebanon and Morocco considering amended legislation to support medical cannabis cultivation and export [16,17,18,19]. 

### 1.3. Russia and Belarus 

Russia and Belarus have a shared history with cannabis dating back to 600 BCE when the Scythians, a nomadic Indo-European group, introduced its use to southeast Russia. Cannabis (mostly *Cannabis sativa*) cultivation in regional territories began in the 8th century [20]. The substance quickly became a popular crop among peasants for propagation and use. In the early 20th century, *Cannabis indica* extract and tinctures were used for medical conditions including pain and insomnia, as well as a laxative when used in liquid form [21]. 

For most people of the Russian Empire, cannabis was not a popular drug especially when compared to alcohol. However, in 1914, shortly before the outbreak of World War I, a prohibition was introduced in the Russian Empire to reduce alcohol consumption [22]. This measure increased the use of drugs such as hashish and anasha (the slang term for hashish and marijuana in some areas of the Russian Empire). Presently, countries now have a zero-tolerance policy for all drugs, including cannabis. While some research on the substance is allowed for medical purposes, there is no education offered about its use at medical universities in Russia or Belarus [23,24]. 

The purpose of this study was to examine whether country of origin affects medical cannabis knowledge, attitudes, and beliefs among foreign medical students studying in Russia and Belarus. We hypothesized that student country of origin is a significant influence.

## 2. Methods

### 2.1. Design, Participants, Procedures 

This cross-sectional study was conducted in 2018–2019 at three Russian and Belarusian medical universities. The study cohort included 549 foreign medical students—47.4% (*n* = 260) from Middle Eastern countries, mostly Egypt, Iran, Syria, and Jordan, and 52.4% (*n* = 289) from India. Among those surveyed, 56.8% (*n* = 312) were female and 43.2% (*n* = 237) male. The mean age of the participants was 21.2 years (SD = 2.8), the median 21.0 years, and 81.2% reported to be religious. For comparison purposes, data were used from Russian and Belarusian origin medical students (RBMS) from the same universities involved (*n* = 796; 55.3% female, 44.7% male; mean age 20.7 years (SD = 2.4); median 21.0 years). For foreign students, the survey was conducted in English.

### 2.2. Instrument and Measures

A data collection instrument was prepared by the Ben Gurion University of the Negev—Regional Alcohol and Drug Abuse Research (RADAR) Center. The United States National Institute on Drug Abuse (NIDA) has given recognition and award to the BGU RADAR Center for its international collaborative research. Questions were prepared in English, translated into Russian, and back translated to English by a multi-national team of researchers to ensure content and vocabulary were appropriate to the students surveyed. The data collection instrument, in addition to demographic questions, included items selected from a literature review of cannabis in educational and clinical settings in domains of usefulness, risks, benefits, treatment, training, and research [25]. The instrument used at the Russian and Belarusian universities went through multiple versions prior to distribution among the students involved. 

To ensure the methods proposed for this research were ethical, appropriate committee approval was received from the Penza State University (Institute of Medicine), Far Eastern Federal University (School of Biomedicine), and Belarus State Medical University. Such research ethics approval was equivalent to established regulations that protect the rights and welfare of human research subjects [26]. No external grant funding was received for the study.

The questionnaire was distributed to medical students in class. This method, rather than an online approach, was chosen to maximize the number of responses in the limited time period allowed for data collection. Frequencies describe respondent characteristics; percentages were used for categorical variables; and, means, medians, and standard deviations were used to address continuous variables. Knowledge, attitude, and belief scores were compared with respondent characteristics across participating universities using the Pearson’s chi-squared test for dichotomous variables, and *t*-test for continuous variables. All statistical analyses were conducted using SPSS, version 25.

## 3. Results

The mean age of participants was 22.7 years among Middle Eastern (MEMS) (SD = 3.0) and 19.8 years among Indian (IMS) participants (SD = 1.8) (*p* < 0.001). Of the MEMS, 24.9% and 12.3% of IMS participants (*p* < 0.001) identified as being secular. No significant differences were found in the rate of personal cannabis use between MEMS and IMS groups (14.3% vs. 17.1%; n.s.). Table 1 provides background characteristics of the survey respondents. Country of origin and gender status did not differentiate Indian and Middle Eastern students. Therefore, the two groups were combined for analysis. 

MEMS and IMS students reported similar attitudes toward the following pain related questions: “I would recommend medical cannabis for patient/client use” (65.8% vs. 69.4%; ns) and “physicians should recommend cannabis as a medical therapy” (57.3% vs. 63.4%; ns). Table 2 shows considerable agreement among the MEMS and IMS participants regarding their beliefs toward medical cannabis effectiveness for pain. 

Irrespective of country of origin, medical students who used cannabis reported more positive attitudes toward the substance for medical purposes. Cannabis users were more likely to recommend MC for patient treatment and support its use, especially for treating arthritis (52.0% vs. 39.3%; *p* < 0.05) and fibromyalgia (45.7% vs. 30.9%; *p* < 0.05). In terms of medical conditions such as chronic pain, cancer, persistent muscle spasm, and multiple sclerosis, no significant differences were found among cannabis users and non-users regarding substance treatment benefits. 

Secular medical students, regardless of country origin, reported more positive attitudes toward cannabis use for medical purposes. Such students were more likely to recommend MC for patient treatment (64.8% vs. 51.0%; *p* < 0.001) and believe it has benefits especially for chronic pain (62.8% vs. 54.8%; *p* < 0.05) and persistent muscle spasms (54.6% vs. 41.9; *p* < 0.001). Regarding other pain-related medical conditions such as arthritis, cancer, fibromyalgia and multiple sclerosis, no significant differences were found among secular and non-secular students regarding cannabis treatment benefits. 

Figure 1 and Figure 2 provide information about MEMS, IMS, and RBMS medical student cannabis attitudes and beliefs. Figure 1 shows that foreign students report more positive attitudes toward MC use than their Russian and Belarusian counterparts. Foreign students, more than those of Russian and Belarusian origin combined, reported more personal cannabis use (15.8% vs. 6.9%; *p* < 0.001), feeling better prepared to answer client questions about MC (60.7% vs. 48.1%; *p* < 0.001), and having received formal education about the substance (41.6% vs. 14.7%; *p* < 0.001). Moreover, foreign students tend to be more supportive of the need for drug policy change regarding cannabis for medical purposes (82.2% vs. 72.2%; *p* < 0.001). 

With respect to MC information sources, foreign medical students were more inclined to use formal sources such as medical literature and classroom lectures (78.1% vs. 68.3%; *p* < 0.001) alongside informal sources such as mass-media and internet (75.2% vs. 36.9%; *p* < 0.001) than their Russian and Belarusian counterparts.

## 4. Discussion

We compared the attitudes, beliefs, and knowledge about MC use for pain relief among foreign medical students attending medical universities in Russia and Belarus where cannabis is banned for recreational and medical purposes. We predicted student attitudes, beliefs, and knowledge about MC would differ based on country of origin status. This was not found to be the case.

Furthermore, present study results evidence no differences among foreign students based on their gender or religious (secular/non-secular) statuses. Students who reported personal cannabis use were more likely to recommend MC for treatment of medical conditions and believe in the benefits of MC for physical and mental health purposes. Regardless of country of origin, foreign students who used cannabis were less likely to believe it poses serious physical or mental health risks. This outcome is consistent with research of medical students elsewhere [27].

Limited research exists about foreign medical student MC knowledge, attitudes, and beliefs, especially those from countries where recreational and medical cannabis use is banned [28]. However, when Indian and Middle Eastern students were combined for analysis purposes because of similar survey responses, we found they had more positive attitudes toward MC than their counterparts from Russia and Belarus. In addition, foreign medical students tend to be more inclined to use both formal and informal sources of information about cannabis and feel more prepared to answer patient MC related questions than those from Russia and Belarus. These results may be linked to cultural and historical reasons. In Russia and Belarus, unlike India and Middle Eastern countries, there has never been large-scale cannabis use. Cannabis use in Russia and Belarus, even when the substance was not banned, was limited to marginal groups, and not widely used like alcohol. Due to national policy, anti-drug propaganda, and the lack of objective information, the status of cannabis use in any form continues to remain mostly negative among Russian and Belarusian people. Nevertheless, this study provides useful information about medical school students based on country of origin that may be applied to curricula and education purposes, especially in the event of possible government change toward cannabis, especially for medicinal purposes.

## 5. Limitations

The primary limitation of this study is that findings are based on a limited number of student respondents surveyed at one point in time at only three universities. In addition, cannabis-related policies and enforcement practices in Russia and Belarus may have influenced student responses about their MC attitudes and beliefs. Factors linked to cannabis policy, religious affiliation, religiosity, and student age should be considered for future research purposes including data analysis methods that go beyond those used for this present study.

## 6. Conclusions 

This study is the first to examine the attitudes, knowledge, and beliefs toward medical cannabis among foreign students from India and Middle Eastern countries studying in Russia and Belarus, two countries who oppose its recreational and medicine use. Indian and Middle Eastern students, as a group, tend to be more supportive of MC than their Russian and Belarusian counterparts. These results may be linked to cultural and historical reasons. This study provides useful information for possible medical and allied health curriculum and education purposes.

## Figures and Tables

**Figure 1 ijerph-18-02137-f001:**
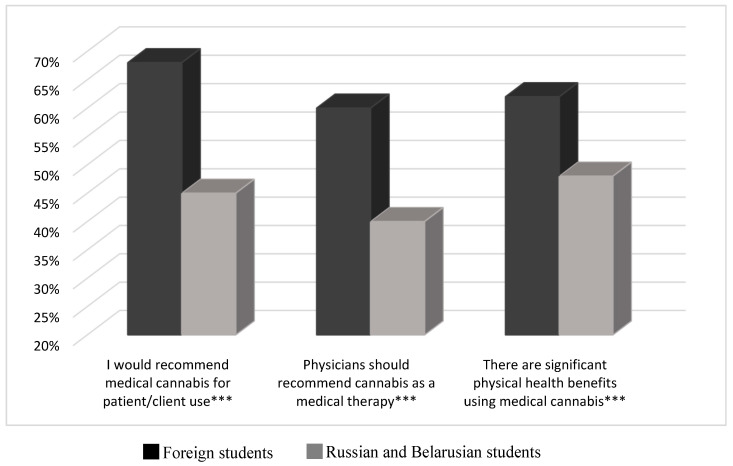
Medical cannabis attitudes among medical students. *** *p* < 0.001 (Chi-square test).

**Figure 2 ijerph-18-02137-f002:**
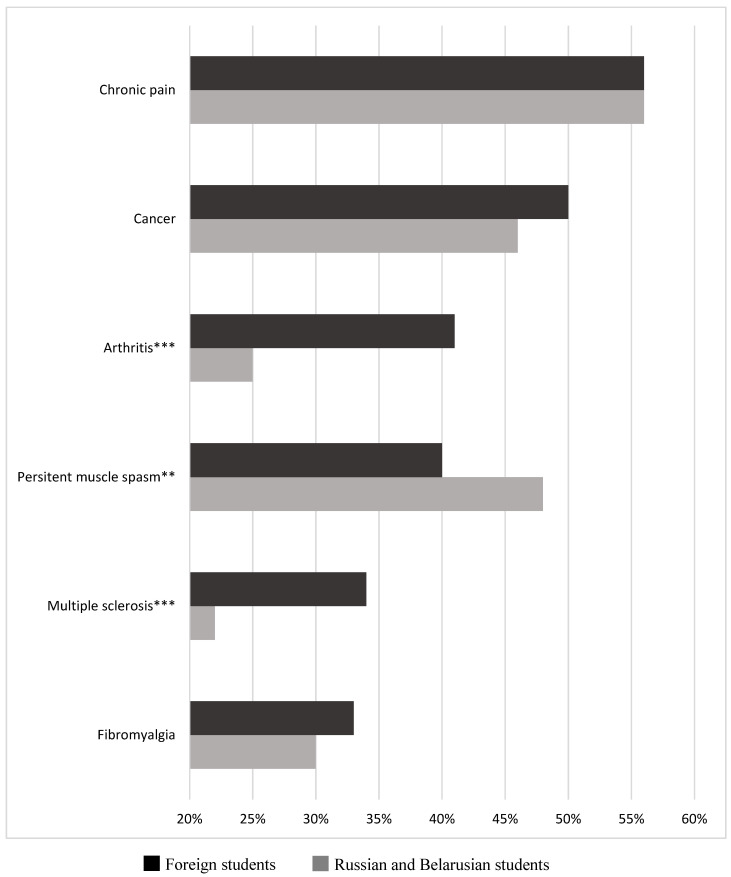
Medical student beliefs about cannabis effectiveness for pain-related conditions, ** *p* < 0.01; *** *p* < 0.001 (chi-square test).

**Table 1 ijerph-18-02137-t001:** Survey respondent background characteristics.

	Middle Eastern Medical Students (*n* = 260)	Indian Medical Students (*n* = 289)	Total (*n* = 549)
Gender, % (*n*)			
Female	57.3 (149)	56.4 (163)	56.8 (312)
Male	42.7 (111)	43.6 (126)	43.2 (237)
Age, mean (SD)	22.7 (3.0) ***	19.8 (1.8) ***	21.2 (2.8)
(Median)	(22.0)	(20.0)	(21.0)
Level of religiosity, % (*n*)	***	***	
Secular	24.9 (61)	12.3 (34)	18.2 (95)
Non secular	75.1 (184)	87.7 (242)	81.8 (426)
Marital status, % (*n*)	***	***	
Married/partner	34.9 (90)	18.8 (54)	26.4 (144)
Other or non-denominational	65.1 (168)	81.3 (234)	73.6 (402)
Prior cannabis use, % (*n*)	14.3 (36)	17.1 (49)	15.8 (85)
Family member cannabis use, % (*n*)	15.0 (38)	12.7 (36)	13.8 (74)
Friend(s) cannabis use, % (*n*)	29.6 (75) ***	43.6 (122) ***	37.0 (197)

*** *p* < 0.001 (*t*-test for age; chi-square test for other variables).

**Table 2 ijerph-18-02137-t002:** Belief: medical cannabis effectiveness for pain-related conditions.

	Middle Eastern Medical Students (*n* = 260)	Indian Medical Students (*n* = 289)	Total (*n* = 549)
Arthritis, % (*n*)	37.5 (92)	44.7 (119)	41.3 (211)
Cancer, % (*n*)	55.2 (139) *	45.6 (119) *	50.3 (258)
Chronic pain, % (*n*)	57.4 (139)	55.1 (145)	56.2 (284)
Fibromyalgia, % (*n*)	33.9 (84)	33.0 (86)	33.4 (170)
Multiple sclerosis, % (*n*)	30.5 (74)	36.9 (96)	33.8 (170)
Persistent muscle spasm, % (*n*)	43.5 (107)	36.3 (97)	39.8 (204)

* *p* < 0.05 (chi-square test).

## Data Availability

The data presented in this study are available on request from the corresponding author. The data are not publicly available due to local restrictions.

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
