# Peer review of "Foreign Medical Students in Eastern Europe: Knowledge, Attitudes and Beliefs about Medical Cannabis for Pain Management"

_ijerph, 2021, doi:10.3390/ijerph18042137_

Round 1
Reviewer 1 Report
This is a well-written paper on an important and interesting topic. I do offer a few suggestions to improve it.
- Why were there universities/populations selected for this study?
- Line 32 define MC when it is used for the first time.
- It is interesting to note the nearly 50/50 split of males and females in medical school. Is this fact for these countries or just for this sample?
- Table 2 needs to be retitled. It has the same title as table 1. I am not clear what table 2 is showing. Lines 133-134 say it is about the belief of MC for pain but it lists other diagnoses. This must be clarified. Additionally, how is table 2 different than figure 2? Figure 2 needs clarification as well.
- Once the tables are clarified the results should be discussed clearly.
Author Response
Dear Reviewer,
Thank you for your response and comments. We took the time to reply to every comment you made. Please see our response in bold under each of your comments.
Comment 1: Why were there universities/populations selected for this study?
Answer 1: Unfortunately, we do not understand the above question.
Comment 2: Line 32 define MC when it is used for the first time.
Answer 2: Thank you for the above comment. We will make the relevant changes in the text.
Comment 3: It is interesting to note the nearly 50/50 split of males and females in medical school. Is this fact for these countries or just for this sample?
Answer 3: In general, women predominate among medical students (including universities in the Middle East and Iran). In this regard, a situation when men and women are equally split is not a typical one. Perhaps this is due to factors that determine education at a foreign university (for example, the reluctance of girls to go to an unfamiliar country).
Comment 4: Table 2 needs to be retitled. It has the same title as table 1. I am not clear what table 2 is showing. Lines 133-134 say it is about the belief of MC for pain but it lists other diagnoses. This must be clarified. Additionally, how is table 2 different than figure 2? Figure 2 needs clarification as well.
Answer 4: This is a mistake in the electronic system for submitting manuscripts to the journal. In the original manuscript, this table has a different name. This situation will be checked upon re-submission of the manuscript.
Comment 5: Once the tables are clarified the results should be discussed clearly.
Answer 5: Unfortunately, we are limited by the size of the text and have to cut things down.
Please find attached the Word document with the correction.
On behalf of all authors, we sincerely thank you again for your comments and we are looking forward to your response.

Reviewer 2 Report
The present article evaluates the knowledge, attitudes and beliefs of students from Middle Eastern countries compared to Russian and Belorussian students towards the use of medical cannabis (MC) for pain management.
There are a number of issues that could also be better addressed:
1. The introduction needs to be reorganized: I suggest starting by describing cannabis (lines 38-44), continuing with the legalization of cannabis for medical purposes (lines 31-37) and ending with the importance of the topic for medical students ( lines 18-30). Moreover, delete paragraphs 1.1, 1,2 and 1,3 and and put the shortened text regarding the history of cannabis directly into the introduction. I suggest focusing on current cannabis use in the countries considered in the study.
2. Methods: Also, this section should be reorganized. Combine the two paragraphs and describe in general the number and the participant. In my opinion, the lines 88-92 and lines 94-95 should be put in results section.
3. lines 100-103 “Questions were prepared in English, translated into Russian, and back translated to English by a multinational team of researchers to ensure content and vocabulary were appropriate to the students surveyed”. Why? Was not it simpler to write in Russian and then translate into English?
4. Add the Russian and Belorussian characteristics in tables 1 and 2.
5. Discussion: I suggest to emphasise the new and important aspects of the study respect both previous authors studies and international literature on the topic.
Some articles to take in consideration:
· Khamenka N et al. Knowledge, Attitudes and Beliefs about Medical Cannabis among the medical students of the Belarus State Medical University. Complement Ther Med. 2021
· Gritsenko V et al. Religion in Russia: Its impact on university student medical cannabis attitudes and beliefs. Complement Ther Med. 2020 Nov;54:102546.
· Jodati AR et al. Students' attitudes and practices towards drug and alcohol use at Tabriz University of Medical Sciences. East Mediterr Health J. 2007 Jul-Aug;13(4):967-71.
Author Response
Dear Reviewer,
Thank you for your response and comments. We took the time to reply to every comment you made. Please see our response in bold under each of your comments.
Comment 1: The introduction needs to be reorganized: I suggest starting by describing cannabis (lines 38-44), continuing with the legalization of cannabis for medical purposes (lines 31-37) and ending with the importance of the topic for medical students ( lines 18-30). Moreover, delete paragraphs 1.1, 1,2 and 1,3 and and put the shortened text regarding the history of cannabis directly into the introduction. I suggest focusing on current cannabis use in the countries considered in the study.
Answer 1: We are grateful to the reviewer for such a suggestion, but in our opinion, this violates the logic of the presentation. We prefer to leave this unchanged, since we believe that our version of the presentation of information in the text of the article is logical and focused on creating a holistic picture of the readers. We sincerely rely on your understanding of our position in regards to our presentation.
Comment 2: Methods: Also, this section should be reorganized. Combine the two paragraphs and describe in general the number and the participant. In my opinion, the lines 88-92 and lines 94-95 should be put in results section.
Answer 2: We are sincerely grateful to the distinguished reviewer for the suggestion, but we assumed that placing this information in "Methods" section is also permissible.
Comment 3: lines 100-103 “Questions were prepared in English, translated into Russian, and back translated to English by a multinational team of researchers to ensure content and vocabulary were appropriate to the students surveyed”. Why? Was not it simpler to write in Russian and then translate into English?
Answer 3: Modified questionnaire, developed by the University of Colorado School of Medicine (ref # 25 in text), was used in the study. Therefore, a translation of the questionnaire into Russian was required.
Comment 4: Add the Russian and Belorussian characteristics in tables 1 and 2.
Answer 4: Main purpose of our article is to show results of comparing students from the Middle East and India. Therefore, information about students from Russia and Belarus will be irrelevant for these tables and are practically not used in the article.
Comment 5: Discussion: I suggest to emphasise the new and important aspects of the study respect both previous authors studies and international literature on the topic. The suggested sources (Khamenka et al. and Gritsenko et al.) are used in the Discussion section Some articles to take in consideration :
- Khamenka N et al. Knowledge, Attitudes and Beliefs about Medical Cannabis among the medical students of the Belarus State Medical University. Complement Ther Med. 2021
- Gritsenko V et al. Religion in Russia: Its impact on university student medical cannabis attitudes and beliefs. Complement Ther Med. 2020 Nov;54:102546.
- Jodati AR et al. Students' attitudes and practices towards drug and alcohol use at Tabriz University of Medical Sciences. East Mediterr Health J. 2007 Jul-Aug;13(4):967-71.
Answer 5: Two of the three authors (Khamenka et al., And Gritsenko et al.) proposed are used in the work (links in text # 28.29)
On behalf of all authors, we sincerely thank you again for your comments and we are looking forward to your response.
Round 2
Reviewer 2 Report
The authors did not amend the Introduction Section. In this regard I believe that presentation of the study is confused. Thus answer of the authors is not acceptable
Author Response
Dear Reviewer,
Our team of authors expresses sincere gratitude to reviewers for their attention to our article and recommendations made.
We agree with the recommendation of the reviewer related to the reorganization of the text of the Introduction. Changes made to the text are highlighted in yellow.
In addition, the text of the article (section "Discussion") is supplemented with important information about the data obtained in the course of the research at the University of Tabriz (the corresponding link # 30, included in References).
Thank you again.
